# Spectroscopic Investigation of the Impact of Cold Plasma Treatment at Atmospheric Pressure on Sucrose and Glucose

**DOI:** 10.3390/foods11182786

**Published:** 2022-09-09

**Authors:** Anna Hauswirth, Robert Köhler, Lars ten Bosch, Georg Avramidis, Christoph Gerhard

**Affiliations:** 1Faculty of Engineering and Health, University of Applied Sciences and Arts, Von-Ossietzky-Straße 99, 37085 Goettingen, Germany; 2Department for Knowledge and Technology Transfer, University of Applied Sciences and Arts, Hohnsen 4, 31134 Hildesheim, Germany; 3Dipartimento di Fisica, Politecnico di Milano, Piazza Leonardo da Vinci 32, 20133 Milan, Italy

**Keywords:** sugars, atmospheric pressure plasma, low temperature plasma, degradation, glucose, sucrose

## Abstract

When exposing food and feedstuff to cold atmospheric pressure plasmas (CAPP), e.g., for decontamination purposes, possible unwanted effects on the contained nutrients might occur. In the present study, we thus concentrated on CAPP-induced degrading effects on different sugars, namely glucose and sucrose. The treatments were performed using admixtures of argon and synthetic air over durations of up to 12min. Continuous degradation of sucrose and glucose was determined using ATR-FTIR and XPS analyses. OH stretching bands showed notable broadening in the ATR-FTIR spectra, which possibly indicates reduced crystallinity of the sugars caused by the CAPP treatment. In the fingerprint regions, most bands, especially the more intense C-O bands, showed decreases in peak heights. In addition, two new bands occurred after CAPP treatment. The bands were detectable in the range between 1800 and $1600\;{\mathrm{cm}}^{-1}$ and potentially can be assigned to C=C and, after comparison with the results of the XPS measurements, O-C=O bindings. The XPS measurements also showed that the O-C=O bonds probably originated from earlier C-O bonds.

## 1. Introduction

Contaminations of food and feedstuff are an ever present problem. The chemical and microbial contamination of food and feedstuff products poses threats such as a shortened shelf life and negative health effects depending on the contamination rates and overall toxicity of the contaminants. In the past, different decontamination strategies were developed to treat contaminated crops [1,2,3,4,5,6].

Recently, the suitability and potential of cold atmospheric pressure plasmas (CAPP) for the decontamination of foods was shown and reported. For example, it was proven that cold plasmas can be used to inhibit the growth of fungi in the vegetative state and to completely inactivate various phytopathogenic fungal species [7,8,9,10,11,12]. Furthermore, it was shown that the degradation of their secondary metabolites, known as mycotoxins, is also achievable [13] as well as the degradation of pesticides [14,15]. Therefore, plasma can play an interesting role in the decontamination of different crops and bulk goods; for example, to enhance storage times. However, the effect of a CAPP treatment on the nutritional value of treated goods can become a limiting factor.

At this point, it is important to thoroughly investigate the impact of a plasma treatment on the nutrients of the treated product. When applying CAPP, aiming at the decontamination of food crops and other nourishing goods, it is vital to understand the occurring degradation effects on toxic contaminants as well as the possible negative effects on contained nutrients. This knowledge can help to customise the application, especially for sensitive ground products. In the course of the presented study the investigation of the impact of plasma on sugar as one of the main nutrients within food was investigated against the background of decontamination. For that reason different sugars were used as substitute for both, nutrients and mycotoxins, due to their overall comparably complex chemical structures.

To investigate possible plasma-induced changes of the structure of the sugars, Fourier transform infrared spectroscopy in the attenuated total reflection mode (ATR-FTIR) was chosen as the main analysis method due to its quick and easy application [16]. Furthermore, this approach facilitates quantitative investigations [17,18,19,20]. Additionally, X-ray photoelectron spectroscopy (XPS) measurements were carried out to reference the ATR-FTIR measurements and to obtain possible additional information.

Until now, only a few studies have been published that deal with the plasma treatment of sugars. However, so far only the effects of plasma on, for example, sugars dissolved in water or fruit juices have been investigated [21,22]. To the best of our knowledge, this is the first time that the plasma-induced effects on sugars in dry conditions were investigated. By performing the following investigations, we aimed to establish and demonstrate the idea of relations between the decontamination of unwanted toxins on the one hand and the possible accompanying and negative reduction in the nutritional value of plasma-treated food products on the other hand.

## 2. Materials and Methods

### 2.1. Sample Preparation

The examined sugars were D(+)-Sucrose (≤99.5%, p.a.) and D(+)-Glucose (p.a., ACS, anhydrous) (Carl Roth GmbH + Co. KG., Karlsruhe, Germany). Figure 1 shows the molecules in Haworth projection with labeling according to Brizuela et al. [23].

Using a vibratory mill (Retsch GmbH, Haan, Germany, type MM400), the sugars were wet-milled to sugar pastes, so-called slurries, for 90 min at a frequency of $20\;\frac{1}{s}$. The dispersing agent was isopropanol and glass beads with a diameter of 1mm to 1.5mm were used as the grinding balls.

The knowledge of the exact sugar content in the slurries was lost when the slurries were separated from the grinding balls. The slurries (S0) were further diluted with isopropanol (S1) for an adequate viscosity to be taken up with a pipette (Eppendorf SE, Hamburg, Germany). Then, 100μL were applied to glass slides (Knittel Glass GmbH, Braunschweig, Germany) and dried on a hotplate at 70°C for about 30s. The sugar masses now on the glass slides—and thus also the approximate sugar content in S1—were then determined with quantitative ATR-FTIR spectroscopy and calibration lines (this is described in more detail in Section 2.3.1). In order to carry out comparable quantitative measurements, the amount of isopropanol that had to be added to S1 in each case was calculated so that the sugar proportions in the glucose and sucrose slurries were approximately the same (S2). In addition, it was decided to further reduce the viscosity of the slurries by using more isopropanol (S3) as this minimises residues of the slurries in the pipette tip and thus also reduces the deviations in the amounts of sugar on the glass slides. Subsequently, 100μL of the resulting slurries was applied to glass slides and dried in layers on a hotplate at 70°C, with each layer left for about 10s. This resulted in layers of sugar with a diameter of approx. 1cm. The amounts of sugars were 1±0.2mg for glucose and 1.1±0.4mg for sucrose. The large deviations in the sugar amounts are also due to the measurement method (quantitative ATR-FTIR spectroscopy, Section 2.3.1).

The next aim was to perform qualitative measurements of plasma-induced chemical alterations via FTIR spectroscopy and XPS. For this purpose, major deviations in the amounts of sugar were accepted. Therefore, the final dilution with isopropanol was dispensed and only 10μL instead of 100μL of the slurries (S2) was dropped onto microscopy cover slips. The subsequent drying on a hotplate at 70°C did not have to be carried out in layers due to the lower isopropanol content. As an advantageous effect, the resulting layers had a significantly smaller diameter of around 5mm and were more homogeneous. These samples were examined with a 3D microscope. Here, a layer thickness of about 120μm was determined at the circumference, whereas the sample thickness was approx. 50μm at its centre. These large deviations can be explained due to formation of coffee ring-like shapes of the samples subsequent to the drying process, induced by surface tensions of the diluted slurries. Subsequently, sugar samples were plasma treated for 1, 2, 4, 8 and 12min in order to gain information on the time evolution and temporal impact of plasma treatment.

It should be noted that even though most foods feature a high water content, dry sugars were investigated in the present work in order to exclude misrepresentations caused by the solution as well as any impact and interacting interdependencies of the dissolution. Hence, the basic effects of the applied plasma treatment on sugar molecules can be identified and evaluated as presented in Section 3.

### 2.2. Plasma Source and Treatment

The plasma source used in our setup was operated at atmospheric pressure. It primarily consisted of a conical hollow electrode (see Figure 2).

At a distance of approx. 15mm to the tip of the hollow electrode, a glass plate was positioned which functioned as a dielectric on the counter electrode. A mixture of 80% argon and 20% synthetic air, adjusted with flow meters (Krohne, Duisburg, Germany, type DK800), was used as the working gas. A pulsed alternating high voltage was applied to the hollow electrode, as a result of which a stationary, thread-like direct plasma was ignited between the hollow electrode and the counter-electrode. This plasma thread consists of many individual filaments. Due to the conical shape of the hollow electrode, the plasma thread is surrounded and stabilised by the gas flow [24].

In the course of the test series, the plasma power was determined by the product of applied voltage and discharge current using an oscilloscope (Agilent Technologies, Santa Clara, CA, USA) [25,26]. The voltage was measured between the high-voltage source and the plasma head using a high-voltage probe (Tektronix, Beaverton, OR, USA, type P6015A). The measured frequency of the AC voltage fluctuated between 8 and 9.5kHz and the voltage was found to be between 8.5 and 10kV. The resulting discharge current was measured at the counter electrode using a wideband current monitor (Pearson electronics, Palo Alto, CA, USA). The power dissipated in the plasma during the test series and it amounted to 10±2W. The voltage and current curves resulting from the measurements at a power of approximately 10W are shown in Figure 3.

### 2.3. Surface Chemistry

The surface chemistry of the sugars before and after plasma treatment was measured using FTIR spectroscopy and XPS. Quantitative measurements of sugar levels before and after plasma treatment were also performed using FTIR spectroscopy.

#### 2.3.1. FTIR Measurements

The plasma-treated sugars were analysed with a commercial FTIR spectrometer (PerkinElmer, Waltham, United States, type Frontier) in the ATR mode. For each measurement, 64 scans were made with a resolution of $4\;{\mathrm{cm}}^{-1}$ in a wavenumber range from 4000 to $500\;{\mathrm{cm}}^{-1}$. For this purpose, an ATR unit (Specac Ltd, Orpington, United Kingdom) which works with a diamond ATR element via single reflection was integrated into the FTIR spectrometer. All spectra were converted to absorbance spectra and automatically ATR corrected and baseline corrected with the program *Spectra* from PerkinElmer. In order to provide quantitative evaluations of the ATR-FTIR spectra, the spectra were also vector normalised using the Euclidean normalisation [16]:(1)$$y^{norm}=\frac{y_{i}}{\sqrt{\sum_{i=1}^{n}{\left(y_{i}\right)}^{2}}}.$$

In the first step of vector normalisation, the mean *y* value is subtracted from the spectrum so that the center of gravity of the spectrum is on the zero line. This causes part of the spectrum to be negative. Therefore, the minimum of the spectrum was then shifted to the zero line, so that the entire spectrum was in the positive range again.

Some studies showed that ATR-FTIR spectroscopy is suitable for the the quantitative analysis of aqueous solutions [17,18,19,20]. Based on this, calibration lines were recorded in a preliminary series of measurements, which established a relationship between the peak height of the sugar band in the ATR-FTIR spectrum and the sugar concentration in distilled water. For this purpose, solutions with concentrations of up to $400\;\frac{\mathrm{mg}}{\mathrm{mL}}$ sugar in distilled water were measured using ATR-FTIR spectroscopy. In addition to the broad water bands, the spectra also showed intense bands in the fingerprint area that were a result of the dissolved sugar. C-O bands of the sugars at around $1054\;{\mathrm{cm}}^{-1}$ were used for evaluation. The peak heights of these bands were averaged and plotted against the associated concentration, resulting in an almost linear relationship. These calibration lines facilitated the determination of unknown sugar concentrations.

Sugars were plasma treated for 1, 2, 4, 8 and 12min. For the quantitative analysis, the plasma-treated sugars as well as the untreated reference samples were dissolved in 10μL of distilled water and dropped onto the ATR crystal so that ATR-FTIR spectra could be recorded. The peak heights of the C-O bands were compared with the previously determined calibration lines to determine the sugars’ mass after the plasma treatment.

In addition, a qualitative examination of the plasma-treated sugars was carried out. Accordingly, the treated sugars on the glass slides were directly pressed onto the ATR crystal to record ATR-FTIR spectra.

#### 2.3.2. XPS Measurements

To determine the chemical binding state, XPS measurements were performed on a PHI 5000 Versa Probe II (ULVAC-PHI, Chigasaki, Japan) using monochromatic Al-Kα radiation with a photon energy of 1486.6eV and a power of 25W. For this purpose, the slurries were dropped onto microscopy cover slips and dried at 70°C on a hotplate. Detailed spectra of carbon (C1s), oxygen (O1s), and nitrogen (N1s) with a pass energy of 46.95eV, a step size of 0.1eV, and a spot size of 100μm were collected at a constant electron take-off angle of 45° for three samples of each variant. The spectrometer was calibrated to the reference lines of copper and gold at 932.62eV and 83.9eV, respectively. The minimum detector resolution measured at the silver peak Ag (3d5/2) was 0.79eV with a pass energy of 46.95eV. During measurement, the temperature was kept constant at room temperature and the base pressure was $2\times 10^{-6}\;\mathrm{Pa}$. In order to avoid charging effects, the measurements were carried out with the neutralisation of sample charging. The structures were fitted by applying Voigt profiles after conducting a Shirley-type baseline subtraction.

## 3. Results

### 3.1. Measurements Via FTIR Spectroscopy

Figure 4 shows the degradation curves of glucose (black) and sucrose (red) caused by the plasma treatment. This degradation can be described by an exponential decay in both cases.

The formula of the degradation curves is in general
(2)$$M\left(t\right)=y_{0}+M_{0}\cdot e^{\frac{-t}{\tau }}.$$

M(t) indicates the mass of the sugar after the plasma treatment time *t*, $M_{0}$ the initial mass, τ the half-life and $y_{0}$ the threshold. The half-lives for glucose and sucrose resulting from the degradation curves can be found in Table 1.

According to these values, the degradation of sucrose seems to progress slightly faster than in the case of glucose. This observation may be due to a slightly higher power during the plasma treatment of sucrose. While the average plasma power measured here was 10W, it was 9.5W for glucose.

In the course of plasma treatment, a colouration of the sugar surfaces was observed in places, which is due to the oxidation products of the sugars. Figure 5 shows the ATR-FTIR spectra of glucose and sucrose, each untreated and 12min plasma-treated.

The range from 1800 to $800\;{\mathrm{cm}}^{-1}$ of the FTIR spectra of untreated glucose and sucrose shows vibrations of the following vertical bonds: Deformation vibrations (δ) of CH${}_{2}$ and OH bonds, wagging vibrations of CH${}_{2}$, rocking vibrations (ρ) of CH, stretching vibrations (ν) of CO and CC and twisting vibrations of CH${}_{2}$ [23,27,28].

The range of the OH stretching vibrations between 3600 and $3000\;{\mathrm{cm}}^{-1}$ as well as some bands in the fingerprint area of the spectra show some broadening of the bands. This indicates an increasing disorder in OH-bonding and possibly a reduction in crystallinity [29,30]. However, since this broad band does not decrease in peak height despite its broadening, it can be said that the plasma treatment results in general increase in the hydroxy groups. In the range between 3600 and $2800\;{\mathrm{cm}}^{-1}$ several bands overlap, which makes it difficult to determine the progress of peak heights. Because of that, bands of the OH deformation, CH rocking and CH${}_{2}$ deformation vibrations in the fingerprint area were used for evaluation. This is shown in Figure 6.

The band at $1203\;{\mathrm{cm}}^{-1}$ of the glucose spectra and the band at $1208\;{\mathrm{cm}}^{-1}$ of the sucrose spectra, respectively, represent OH peaks. According to Brizuela et al., this can be assigned to the O43-H deformation mode in sucrose [23]. Because this bond belongs to the glucose residue and the FTIR spectra of glucose also shows a similarly intense band in this region, it was assumed that this was the same vibration. This assumption was confirmed by the band assignment provided by Ibrahim et al. [27]. Figure 6a shows the peak heights of this band for glucose (black) and sucrose (red) as a function of the plasma treatment time. While the peak height of this band generally decreases for sucrose, it increases for glucose. However, not all bands of OH vibrations showed this behaviour. For example, the O39-H deformation bands at $1386\;{\mathrm{cm}}^{-1}$ (sucrose) and $1381\;{\mathrm{cm}}^{-1}$ (glucose), respectively, showed a decrease in peak height (see Appendix A).

Figure 6b shows the peak heights of the CH rocking bands at about $1280\;{\mathrm{cm}}^{-1}$. According to Brizuela et al., this band was assigned as the rocking vibration of C32-H [23]. These peak heights for glucose show a course similar to the bands of the OH deformation vibrations, while for sucrose, the peak height of the untreated reference sample is significantly higher, resulting in a stronger decrease.

The bands of CH${}_{2}$ deformation vibration in the range of $1460\;{\mathrm{cm}}^{-1}$ show a rapid asymptotic decrease in the FTIR spectra of both sugars in the course of the plasma treatment (Figure 6c). Brizuela et al. assigned the CH${}_{2}$ deformation mode of C37 to this wavenumber [23]. Independently of this, a few other publications also assigned this band in the FTIR spectra of glucose to a CH${}_{2}$ deformation mode [27,31,32,33]. Because glucose only has one CH${}_{2}$ bond, this vibration can also be assigned to C37.

In Figure 5, a general decrease in the more intense bands in the fingerprint areas can be seen for both sucrose and glucose. In addition to the already mentioned deformation and rocking vibrations of some OH, CH and CH${}_{2}$ bands, these also contain stretching vibrations of the C-O and C-C bands. Figure 7 shows the courses of the peak heights of some C-O (a) and C-C (b) stretching vibrations.

Most of the bands in the fingerprint areas show almost asymptotic decreases in peak heights. Such a course is also shown by the bands of the C28-O34 stretching mode at $994\;{\mathrm{cm}}^{-1}$ in the FTIR spectrum of glucose and at $989\;{\mathrm{cm}}^{-1}$ in the sucrose spectrum, respectively. The assignments of the bands of sucrose were made according to Brizuela et al. [23]. The same assignments were used for the band of glucose at slightly higher wavenumbers. Ibrahim et al. supported this classification, since they also assigned the band to a C-O stretching vibration [27]. The bands at $920\;{\mathrm{cm}}^{-1}$ in the sucrose spectrum and at $917\;{\mathrm{cm}}^{-1}$ in the spectrum of glucose, respectively, are due to C31-C32 stretching vibrations [23,31]. For sucrose, this band shows an asymptotic decrease, while for glucose it is significantly less intense and shows a comparatively slight decrease.

Figure 5 shows a significant increase in peak height at $1077\;{\mathrm{cm}}^{-1}$ in the spectrum of plasma-treated glucose. The assignment of this band is not entirely clear. According to Brizuela et al., the spectrum of sucrose shows the C24-O23 stretching band at a slightly higher wavenumber and that of C31-C37 at a slightly lower wavenumber [23]. Other studies instead assigned this band to C-O stretching [28,32,33].

Figure 8 shows the peak heights of the two bands formed by the plasma treatment in the range between 1800 and $1600\;{\mathrm{cm}}^{-1}$.

The peak heights of the bands at $1733\;{\mathrm{cm}}^{-1}$ of both sugars show the greatest increase within the first minute of plasma treatment. While the peak height of glucose continuous to increase slightly after 2min, the peak height of sucrose decreases between 2 and 8min before remaining almost constant thereafter.

The two bands at 1643 and $1645\;{\mathrm{cm}}^{-1}$ feature roughly the same increasing trend in peak heights, except that those of sucrose are slightly higher than those of glucose in the first two minutes of plasma treatment. These bands are attributed to either the OH deformation modes of water molecules or C=N or C=C stretching vibrations in the aromatic region [34,35]. The latter seems to be the most likely in this context and could indicate degradation products of the sugars. The more intense bands at $1733\;{\mathrm{cm}}^{-1}$ are assigned to C=O stretching vibration and thus indicate a functional carbonyl group [16,34].

### 3.2. Measurements Via XPS

Figure 9 shows the mean values of the C1s peaks from the measured XPS spectra of glucose (top) and sucrose (bottom), each of which were untreated (left) and 12min plasma-treated (right).

The carbon peak can be subdivided into four areas. The maximum at 286.6eV can be assigned to C-O bonds; the spectra were shifted to this peak [36]. The peaks which shifted to higher binding energies at 288 and 289.3eV indicated O-C-O and O-C=O bonds. The shoulder at 285eV may be caused by C-C bonds, indicating atmospheric contamination of the sugar. The O-C=O peaks are faintly visible in the spectra of untreated sugars; the peak becomes significantly more intense as a result of plasma treatment.

In Figure 10, the peak heights of the bands formed during plasma treatment at $1733\;{\mathrm{cm}}^{-1}$ as determined by FTIR spectroscopy are compared to the atomic concentrations of O-C=O measured by XPS.

A very good agreement between the courses can be noticed. When plotting the FTIR measurements versus the XPS measurements, a linear relationship can be seen. Taking the error bars (instrumental weights) into account, the values for glucose correlate with an R-squared value of 0.9983, for sucrose the R-squared value is slightly lower at 0.9297. This indicates that the newly formed bands in the FTIR spectra at $1733\;{\mathrm{cm}}^{-1}$ are the same O-C=O bonds as measured using XPS.

The comparison of the atomic concentrations of the C-O bands and those of the O-C=O bands revealed that the courses of these two bands are almost axisymmetric, with the atomic concentrations of the O-C=O bonds being significantly less intense than those of the C-O bonds. To illustrate this, the atomic concentrations of these bonds are presented in stacked bar charts in Figure 11.

It can be seen that the sum of the C-O and O-C=O atomic concentrations remained almost constant over the plasma treatment time. The mean value of this concentration is shown as a blue dashed line. It amounts to 34.97±0.5% for glucose, while for sucrose it is slightly higher at 36.80±0.7%. This potentially indicates that the O-C=O bond is formed from a C-O bond by attaching another oxygen atom, most likely originating from the used plasma process gas with a double bond.

## 4. Discussion

The peak heights of most OH and CH bands and all CH${}_{2}$ bands, including those on the fructose ring of sucrose, show an almost asymptotic decrease in the course of plasma treatment as observed via ATR-FTIR spectroscopy. A decrease in the peak height of a band might indicate the opening of the associated bond; the detected C-C band shown was found within the glucopyranose ring, and a decrease in its peak height may thus indicate an opening of the ring [37]. Another reason for the decrease in peak heights in the FTIR spectra may be the impeded movement of bonds, for example due to an increased degree of crosslinking [38,39].

In the fingerprint area, only the CH bands shown in our study, the OH band of the CH${}_{2}$ group of glucose as well as the band at $1077\;{\mathrm{cm}}^{-1}$ (see Appendix A) feature an increase in peak heights. The increase in the peak heights of some bands may indicate degradation products whose spectra overlap with those of the less plasma-affected sugar. This could lead to altered peak heights and the broadening of some bands. Unlike a few decreasing OH bands, the bands mentioned show notable band broadening. The spectra of sucrose also show a broadening of these bands. Another possible reason for this observation is that the plasma treatment results in the changing of molecules near the widened bands, resulting in changes of the required vibrational energies and broadening of the bands. The possible increasing disorder and reduction in crystallinity, which has already been described, could also be a cause of the broadening of the bands [29,30].

Since the plasma was ignited in an air environment and due to the fact that synthetic air was added to the working gas, there were more reactive oxygen species available in the plasma, causing oxidation of the sugar surfaces. Both the XPS and FTIR spectroscopic results indicate that oxygen species attach to carbon atoms with a double bond, leading to the formation of carbonyl functional groups. When the ring is closed, this is only possible at C37 in the case of glucose, with the two hydrogen atoms being split off. This creates the uronic acid of glucose, glucuronic acid, whose degradation products include reductic acid and furfural [40,41]. When the ring structure is open, there are additional possibilities for the formation of O-C=O compounds. The processes of hydrolysis and oxidation of sugars produce, among other things, various carboxylic acids, which also show intense bands of of C=O stretching frequencies in the range between 1620 and $1750\;{\mathrm{cm}}^{-1}$ [21,42]. Therefore, the newly formed bands in the FTIR spectra in this range could also be due to some of these acids.

As the results showed, the largest changes in the FTIR spectra occurred within the first minute of plasma treatment. Accordingly, a further study with shorter treatment durations could provide information on the exact course of the peak heights, whereby the use of a different plasma source would probably make sense. The suspected reduction in crystallinity due to the plasma treatment may be clarified by an analysis using X-ray diffractometry. Optical emission spectroscopy during plasma treatment could provide more detailed information about possible degradation products as well as a correlation of the peaks in the emission spectra with those in the ATR-FTIR spectra.

Furthermore, using quantitative ATR-FTIR spectroscopy, a degradation of the sugars by the plasma treatment could be determined. The degradation of both sugars followed an exponential course. Thus, sugars displayed similar degradation effects during plasma treatment as, for example, mycotoxins [13]. The half-lives of the degradation curves suggest that sucrose (τ≈3.22min) degrades slightly faster than glucose (τ≈4.31min), which is probably due to the slightly higher average plasma power during the treatments. Nonetheless, it is clear that both treated sugars in this experiment degrade much slower compared to mycotoxins such as Fumonisin B1 (τ≈1.9s) or Sterigmatocystin (τ≈5.0s) [13]. This outcome seems particularly interesting as the sugar samples were treated at comparable power densities to the mycotoxin samples depicted in [13].

## 5. Conclusions

Both monosaccharide glucose and disaccharide sucrose were plasma treated with cold atmospheric pressure plasma (CAPP) using a gas mixture of argon and synthetic air for up to 12min. It can be concluded that a plasma-based decontamination method, as for example the CAPP treatment used here, can be applied to sugar-containing foods without the risk of degrading the contained sugars substantially—providing a short treatment duration. To validate the suitability of CAPP as a safe food treatment method for decontamination applications in more detail, it is necessary to assess its impact on further nutrient groups such as fatty acids, vitamins, etc., as well as their subgroups.

## Figures and Tables

**Figure 1 foods-11-02786-f001:**
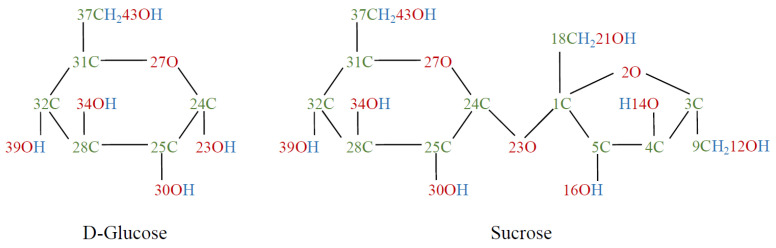
Sugar molecules of D-glucose (**left**) and sucrose (**right**) in Haworth projection.

**Figure 2 foods-11-02786-f002:**
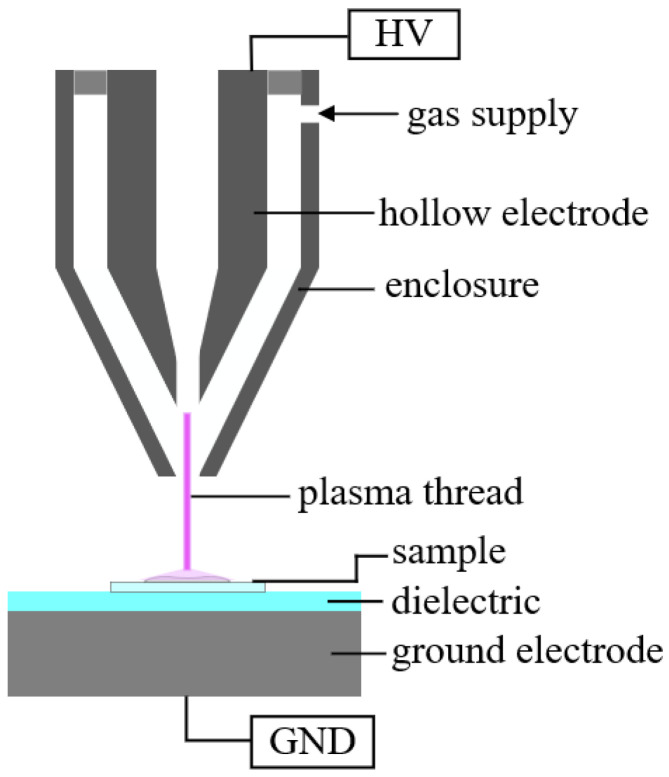
Schematic representation of the structure of the used plasma source.

**Figure 3 foods-11-02786-f003:**
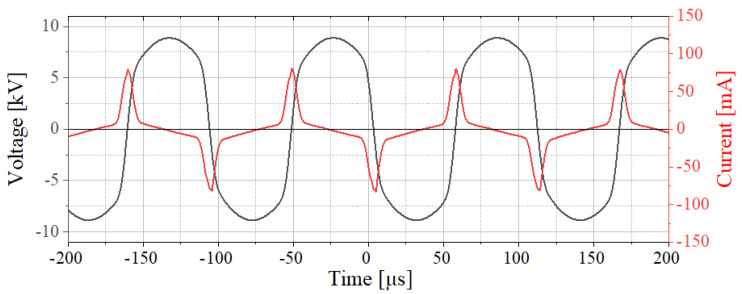
U–I–characteristics of the plasma source at a power of approx. 10W.

**Figure 4 foods-11-02786-f004:**
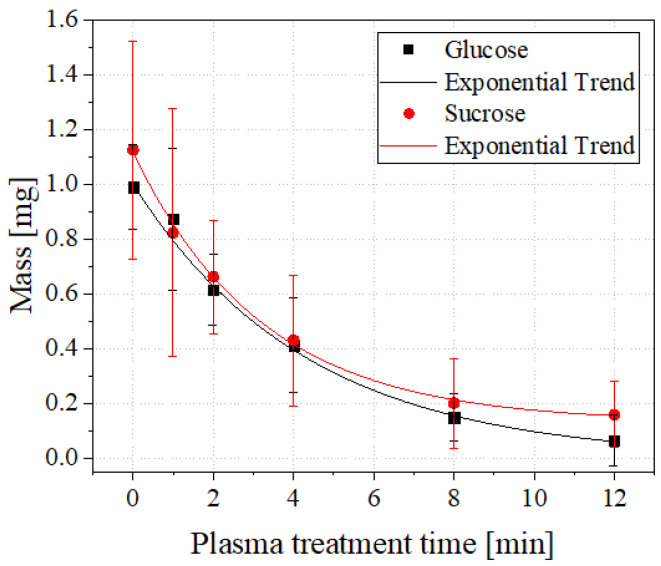
Curves of plasma-induced degradation of glucose (black) and sucrose (red). The degradation of both sugars follow exponential courses.

**Figure 5 foods-11-02786-f005:**
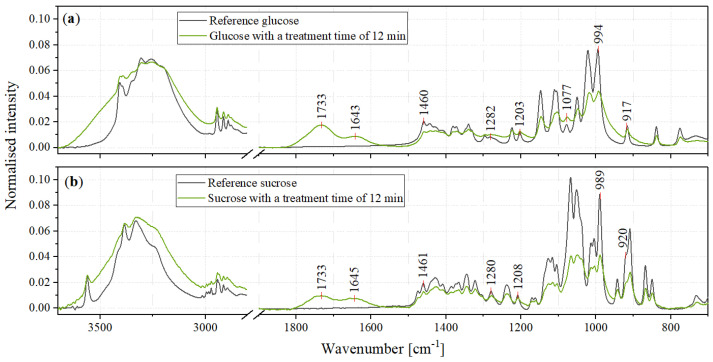
FTIR spectra of untreated (black) and 12min plasma-treated (green) sugars: (**a**) shows this for glucose and (**b**) for sucrose.

**Figure 6 foods-11-02786-f006:**
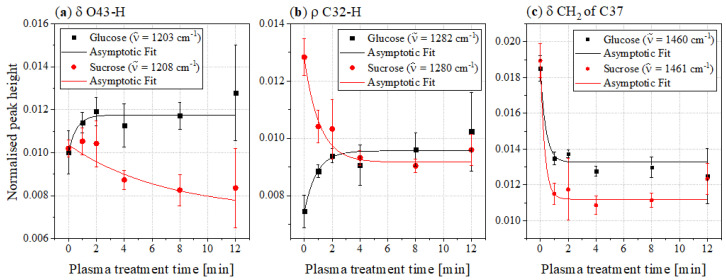
Peak heights of the (**a**) δ OH, (**b**) ρ CH and (**c**) δ CH${}_{2}$ bands in the FTIR spectra of glucose (black) and sucrose (red) as a function of the plasma treatment time.

**Figure 7 foods-11-02786-f007:**
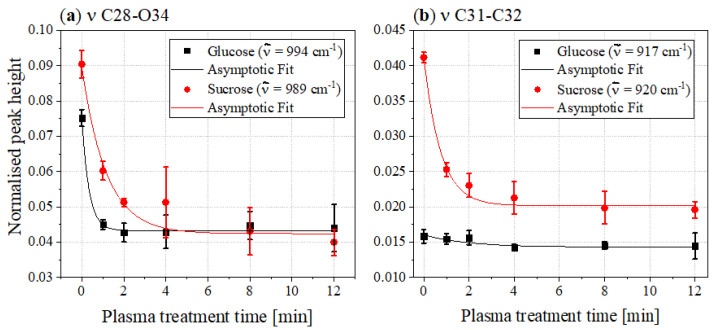
Peak heights of the (**a**) νCO bands and (**b**) νCC bands of the FTIR spectra of glucose (black) and sucrose (red) as a function of the plasma treatment time.

**Figure 8 foods-11-02786-f008:**
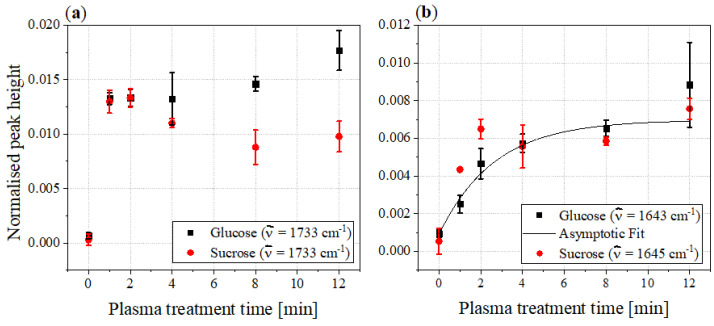
Peak heights of the bands newly formed by the plasma treatment in the FTIR spectra of glucose (black) and sucrose (red) as functions of the plasma treatment time: (**a**) shows the peak heights of the bands at $1733\;{\mathrm{cm}}^{-1}$; (**b**) shows the peak heights of the bands at $1643\;{\mathrm{cm}}^{-1}$ and $1643\;{\mathrm{cm}}^{-1}$, respectively.

**Figure 9 foods-11-02786-f009:**
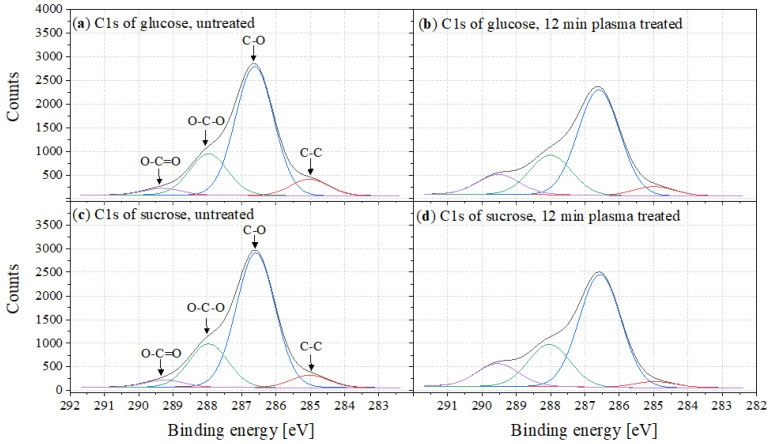
C1s peak of (**a**) untreated glucose, (**b**) 12min plasma-treated glucose, (**c**) untreated sucrose and (**d**) 12min plasma-treated sucrose.

**Figure 10 foods-11-02786-f010:**
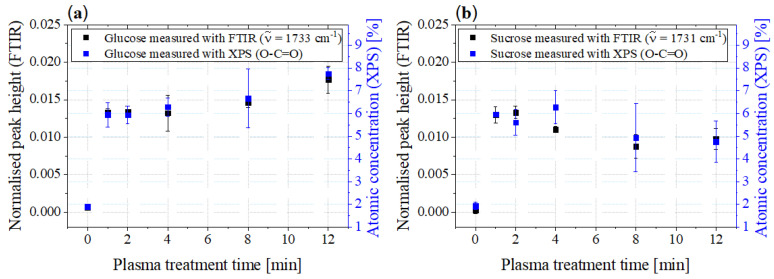
Comparison of the peak height courses of the newly formed bands of the carbonyl compounds in the FTIR spectra with the atomic concentrations of the O-C=O bonds of (**a**) glucose and (**b**) sucrose determined via XPS.

**Figure 11 foods-11-02786-f011:**
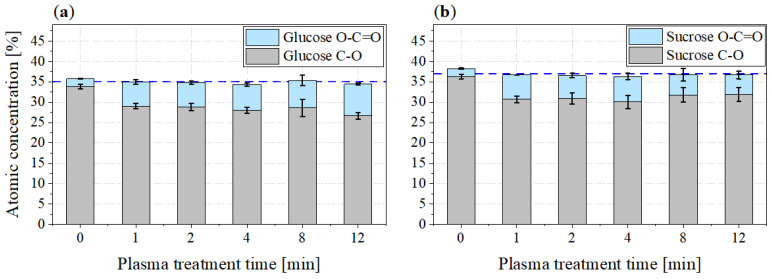
Stacked bar charts of the atomic concentrations of C-O bonds (grey) and O-C=O bonds (blue) of (**a**) glucose and (**b**) sucrose.

**Table 1 foods-11-02786-t001:** Half-lives of glucose and sucrose.

Sugar	τ [min]
Glucose	4.31±0.49
Sucrose	3.22±0.18

## Data Availability

The data presented in this study are available on request from the corresponding author. The data are not publicly available due to the file sizes of the raw data.

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
