# Peer review of "Spectroscopic Investigation of the Impact of Cold Plasma Treatment at Atmospheric Pressure on Sucrose and Glucose"

_foods, 2022, doi:10.3390/foods11182786_

Round 1

Reviewer 1 Report

Dear Authors,

this research brings an interesting topic of plasma interactions with most common food ingredients, and the overall idea is significant. However, it seems that the main goal of this research was driven by analytical equipment that was available to you. My main concern is why did you only use plasma treatment on solid sugars? Most of the food is high in water content, so it is more appropriate to test plasma interactions on dissolved sugars. Also, sugars in combination with microorganisms have protective effects which make plasma less effective as a decontamination technique, so it will be interesting to include microorganisms in these dried sugar slides.

This paper is well written, but it lacks a deeper explanation of the used methodology as well as poor experimental setup description and plasma characterization. 

Author Response

Please find our reply in the attached file...

Reviewer 2 Report

The manuscript presents a detailed spectroscopic overview of the application of cold plasma on sucrose and glucose. Noteworthy is a fairly brief presentation with the latest references. Incredibly, so many new references have been used. Therefore, it is worth accepting this manuscript for publication to spread the use of this technique in food processing. However, I would suggest making some improvements to improve this article.

1.   In the first half of the introduction section authors only focused on the effect of cold plasma on contamination, but this article is related to the impact of cold plasma on sucrose and glucose. So, the authors should revise the introduction section according to the study's aim and focus on clod plasma effects of sugars.

2.   What is the plasma treatment time for the Glucose and sucrose samples?

3.   The authors should explain the treatment time in the methodology section.  

4.   Also need to explain the FTIR section in more detail, especially the 800-1800 cm-1 section for glucose and sucrose.

Author Response

(The authors gave the same response as above.)

Reviewer 3 Report

1. The need to compare this study with mycotoxin degradation is not justified and should be removed. 

2. No details on the plasma species created by this gas system has been shown. All changes discussed is highly dependent on the gas species created during the treatment. Please add proper OES or gas species characterization section. Also add this detail to support the discussion section.

3. Provide the electrical characteristics graph for both glucose and sucrose treatment, since the power were different and it is suggested to be the reason for different half-life.

4. Provide supplementary data to substantiate line 204-206.

Minor Language/spelling corrections in line- 2, 4, 186 and rewrite 22-24. 

Author Response

(The authors gave the same response as above.)

Round 2

Reviewer 1 Report

Thank you for the detailed explanation.